# Translocation of Insecticidal Bt Protein in Transgrafted Plants

**DOI:** 10.3390/biotech14030064

**Published:** 2025-08-25

**Authors:** Arisa Ando, Hitomi Ohkubo, Hisae Maki, Takumi Nishiuchi, Takumi Ogawa, Tomofumi Mochizuki, Daisaku Ohta, Hiroaki Kodama, Taira Miyahara

**Affiliations:** 1Graduate School of Horticulture, Chiba University, 1-33 Yayoi-cho, Inage-ku, Chiba 263-8522, Japankodama@faculty.chiba-u.jp (H.K.); 2Department of Food and Nutrition, Junior College of Kagawa Nutrition University, Toshima-ku, Tokyo 170-8481, Japan; mhisae@eiyo.ac.jp; 3Division of Biological Science and Technology, Graduate School of Natural Science and Technology, Kanazawa University, Kakuma-machi, Kanazawa 920-1192, Japan; takumin@takuminishiuchi.com; 4Division of Integrated Omics Research, Bioscience Core Facility, Research Center for Experimental Modeling of Human Disease, Kanazawa University, Kanazawa 920-8640, Japan; 5Graduate School of Agriculture, Osaka Metropolitan University, 1-1 Gakuen-cho, Naka-ku, Sakai 599-8531, Japan; ogawat@omu.ac.jp (T.O.); tomochi@omu.ac.jp (T.M.); ohtad@omu.ac.jp (D.O.)

**Keywords:** *Bacillus thuringiensis*, Cry1Ab, genetically modified plants, grafting

## Abstract

Transgrafting constitutes a technique involving the integration of genetically modified (GM) and non-GM plant organisms. Typically, edible components derived from non-GM scions are categorized as non-GM food products, attributed to the absence of exogenous genetic material within their respective genomes. Non-GM food status could be compromised if proteins translocated across the graft interface. We investigated the movement of insecticidal *Bacillus thuringiensis* (Bt) crystal proteins, widely utilized in GM crop species. Tobacco plants engineered to express the Cry1Ab gene exhibited trace levels of Cry1Ab protein accumulation. In transgrafted plants, translocated Cry1Ab protein originating from GM rootstocks was detectable within scion foliar tissues but not within the seeds obtained from the non-GM scion. This result unequivocally demonstrates the capacity for Bt protein translocation from rootstocks to scions yet indicates a constrained distribution confined to scion tissues relatively close to the graft junction. While regulatory considerations necessitate a thorough appraisal of potential risks associated with Bt proteins, the results shown here facilitate the commercialization of the edible components as non-GM food products.

## 1. Introduction

Transgrafting represents a specialized grafting methodology that generates a composite plant organism integrating genetically modified (GM) and non-GM tissues [1]. This chimeric architecture permits the non-GM component to acquire advantageous characteristics from the GM plant body, including augmented resistance to abiotic and/or biotic stressors and enhanced productivity, without requiring genomic alterations within the non-GM portion. A remarkable prospective application of transgrafting entails the deployment of GM rootstocks engineered to express *Bacillus thuringiensis* crystal proteins (“Bt proteins”), which confer protection against insect herbivory [2]. GM crops expressing Bt proteins increased the crop yields, reduced pesticide consumption [3], and consequently enhanced the profitability of farmers [4]. The transgrafted plants are thus anticipated to manifest enhanced resistance to herbivorous insects, thereby promoting consistent fruit and seed yields from the non-GM scion.

In 1994, the first commercially available food product, a transgenic “Flavour Saver tomato”, was approved in the United States. Subsequently, in 2003, the Codex Alimentarius established fundamental regulatory frameworks for risk assessment of GM foods. These regulations are essential for the commercialization of GM foods in the majority of countries [5]. In many countries, there is a regulatory approval process for GM crops. Transgene-free plant parts from transgrafted plants might not be subjected to the same regulatory approval process, provided they did not contain translocated protein corresponding to the transgene. However, protein translocation beyond the graft junction has been demonstrated in the transgrafted plants with the rootstocks producing luciferase (LUC) [6], green fluorescent protein (GFP) [7], and chimeric proteins with GFP [8]. While Bt protein translocation has been reported in transgrafted cotton [9] and poplar [10,11], these cases involved nonfood crops or protein-devoid products (cottonseed oil). Thus, no prior evidence existed for rootstock Bt protein translocation into scion fruits or seeds. Here, to elucidate the translocation of Bt proteins into the fruits and seeds of transgrafted plants, the *Cry1Ab* gene was expressed in a model tobacco plant. The donor organism of *Cry1Ab* is *B. thuringiensis* subspecies *kurstaki.* The regulatory approvals for use of Cry1Ab protein in GM food and feed have been issued in 18 countries, as reported [12]. The GM rootstock accumulated modest Cry1Ab levels, which translocated into scion leaves. Remarkably, no Cry1Ab protein was detected in seeds from the non-GM scion. We discuss the potential risks of NEP translocation into edible transgrafted plant portions while affirming the ongoing necessity of NEP risk assessments.

## 2. Materials and Methods

### 2.1. Generation of Plasmids and Transgenic Plants

An actively insecticidal, truncated 648-amino-acid Cry1Ab peptide-encoding nucleotide sequence [13] was optimized for tobacco codon usage (Eurofins Genomics, Tokyo, Japan) and deposited in GenBank (accession LC790689). The predicted amino acid sequence matched that of a Cry1Ab protein previously reported [13]. This fragment, synthesized by Eurofins Genomics, was inserted into pBI121 to replace the β-glucuronidase gene. The resulting pBI-Bt plasmid (Figure 1A) was introduced into *Agrobacterium tumefaciens* LBA4404 for *Nicotiana tabacum* cv. SR1 transformation via the leaf disk method [14]. Two T0 transgenic lines, Bt39 and Bt44, were regenerated. T1 seedlings, derived from self-pollinated T0 plants, were genotyped for T-DNA. Total DNA from T1 Bt01 seedling leaves was extracted [15], and the transgene amplified using primers Bt-Fw (5′-CAGATCCGGCTTTTCCAATA-3′) and Bt-Rv (5′-TTGTGGCGCTAAAATTACCC-3′). The 356 bp amplicon was visualized by gel electrophoresis.

### 2.2. Experimental Transgrafting Procedures

Wild-type (WT) and two Bt line seeds were germinated aseptically, and 3-week-old seedlings were subsequently planted in soil. T1 Bt seedlings underwent genotyping as previously detailed. One month postplanting, Bt rootstocks were grafted with WT scions according to Kodama et al., (2022) [16]. A cohort of 11 independent transgrafted plants was employed for all downstream analyses (Table 1). Protein extraction utilized the two scion leaves immediately adjacent to the graft junction. The second leaf’s distance from the junction ranged from 1 to 6.7 cm (Appendix A). Ungrafted WT and Bt plant leaves were also sampled as controls. All harvested leaves were promptly flash-frozen in liquid nitrogen and stored at −80 °C. Furthermore, six transgrafted plants were self-pollinated to yield seeds, which were subsequently stored desiccated at 4 °C.

### 2.3. Leaf-Specific Cry1Ab Transcript Analysis

For Cry1Ab transcript detection, total RNA was isolated from leaves of four distinct transgenic plants (two Bt39 and two Bt44 lines; Appendix A), and from leaf tissues of the non-GM scion of two independent transgrafted (WT/Bt44) plants (Figure 1C). RNA extraction was performed using a FavorPrep Plant Total RNA Mini Kit (Favorgen Biotech Corp., Ping Tung, Taiwan), with subsequent DNase I treatment (Nippongene, Tokyo, Japan). cDNA synthesis was carried out using Primescript RT reagent (TaKaRa Bio Inc., Kusatsu, Shiga, Japan). Subsequent PCR amplification utilized GoTaq Hot Start Colorless Master Mix (Promega Co., Madison, WI, USA). Amplification of the *Cry1Ab* gene employed the primer described above Bt-Fw and Bt-Rv. The PCR cycling conditions were: initial denaturation at 94 °C for 2 min; 30 cycles of 94 °C for 30 s, 55 °C for 30 s, and 72 °C for 50 s. Visualization of the 356 bp *Cry1Ab* amplicon was achieved via gel electrophoresis. Tobacco *actin* genes served as positive controls, amplified concurrently using specific primers: Actin-F2, 5′-GCACCTCTTAACCCGAAGGC-3′ and Actin-R2, 5′-TGCCTGCAGCTTCCATTCCA-3′.

### 2.4. Cry1Ab Protein Immunoprecipitation

For protein extraction, leaves and seeds were homogenized with an extraction buffer containing 200 mM Tris-HCl (pH 8.0), 0.5 mM ethylenediaminetetraacetic acid, and 10 mM 2-mercaptoethanol. The total protein obtained was subsequently precipitated using ammonium sulfate, then redissolved in 50 mM Tris-HCl (pH 8.0). This solution was desalted via a PD-10 column (Global Life Sciences Technologies Japan, Tokyo, Japan), pre-equilibrated with 50 mM Tris-HCl (pH 8.0). The recombinant Cry1Ab protein from the desalted total protein fraction was specifically immunoprecipitated utilizing an anti-Cry1Ab polyclonal antibody (ab51586, Abcam, Cambridge, UK) in conjunction with a Capturem IP and Co-IP Kit (Takara Bio Inc., Kusatsu, Shiga, Japan).

### 2.5. Liquid Chromatography–Tandem Mass Spectrometry (LC–MS/MS) Analysis

#### 2.5.1. Protein Digestion and Peptide Preparation

Protein solutions (2.5 μg) were subjected to SpeedVac drying and subsequently reconstituted in 50 mM triethylammonium bicarbonate buffer containing 6 M urea. Proteins were then reduced with 5 mM tris (2-carboxyethyl) phosphine for 30 min at 37 °C in the dark, followed by alkylation with 24 mM iodoacetamide for 30 min at room temperature, also in the dark. The alkylated proteins underwent digestion with 250 ng of trypsin (Promega Co., Madison, WI, USA) at 37 °C for 16 h. Digested peptides were purified using a strong cation exchange capillary column (GL Sciences Inc., Tokyo, Japan), adhering strictly to the manufacturer’s protocol. Subsequently, peptides were desalted using a stage tip (Thermo Pierce, Tokyo, Japan; #84850) and eluted with 70% acetonitrile (ACN). The eluted peptides were then vacuum centrifuged to eliminate residual solvent and solubilized in 5% ACN with 0.1% trifluoroacetic acid.

#### 2.5.2. LC–MS/MS Analysis and Identification of Peptide Sequences

Purified peptides were loaded to an Aurora column (25 cm × 75 μm ID, 1.6 mm C18; IonOptics, Fitzroy, Austria) and separated via a linear 0–40% ACN gradient in 0.1% formic acid, at a flow rate of 300 nL min^−1^. Peptide ions underwent identification using an Orbitrap QE plus MS (Thermo Fisher Scientific, Waltham, MA, USA) operating in a data-dependent acquisition mode, as implemented in Xcalibur version 4.4 (Thermo Fisher Scientific, San Jose, CA, USA). Full-scan mass spectra were acquired via MS across the range of 375 to 1500 *m*/*z*, with a resolution of 70,000 *m*/*z*. All raw data files were subsequently processed utilizing PEAKS studio 10 (Bioinformatics Solutions Inc., Waterloo, Ontario, Canada). For de novo peptide identification, a precursor mass tolerance of 10 ppm, a fragment ion mass tolerance of 0.02 Da, and strict trypsin specificity (permitting up to two missed cleavages) were applied. Finally, peptide sequences were mapped to protein sequences of Cry1Ab (Uniprot accession: LC790689) and *N. tabacum* proteins obtained from the UniProt database, including approximately 200 post-translational modifications, using PTM mode.

## 3. Results

### 3.1. Generation of Cry1Ab-Expressing Transgenic Tobacco

Cry1Ab protein, a Bt toxin, is approved for use in multiple GM crops, with its food safety extensively validated [12]. The *Cry1Ab* gene, originating from *B. thuringiensis* subsp. *kurstaki*, confers robust resistance against lepidopteran pests. For this study, we selected two regenerated tobacco lines, Bt39 and Bt44, as transgenic carriers of the *Cry1Ab* gene. Confirmation of T-DNA presence and *Cry1Ab* gene expression in these lines was achieved through PCR amplification of *Cry1Ab* sequences (Appendix A). Western blotting of total protein from Bt44 leaves, using an anti-Cry1Ab antibody, revealed no discernible cross-reactive proteins (Appendix A). This observation strongly suggests that Cry1Ab protein accumulates at low levels in the Bt44 line, aligning with previous reports of low-level *Cry1Ab* expression in various regulatory submissions [12].

### 3.2. Cry1Ab Protein Detection in Transgrafted Non-GM Scion Leaves

WT scions were grafted onto Bt44 rootstocks (Figure 1B). Given the low Cry1Ab protein levels in the Bt44 rootstock, we employed immunoprecipitation for protein concentration. Subsequent LC–MS/MS analysis of immunoprecipitated leaf proteins from WT, Bt44, and transgrafted plants revealed the presence of Cry1Ab protein in the non-GM scion leaves of transgrafted plants (Table 1). Remarkably, four out of five independent transgrafted samples contained Cry1Ab peptides (Table 1). The specificity of these detected peptides to Cry1Ab protein was confirmed by the absence of identical amino acid sequences in the tobacco protein database. Moreover, our RT-PCR analysis detected no *Cry1Ab* mRNA in the non-GM scion of transgrafted samples (Figure 1C), strongly indicating that the observed Cry1Ab peptides originated from proteins translocated from the rootstock. These findings lead us to conclude that Cry1Ab protein translocation from rootstock-to-scion leaves is a frequent event, aligning with observations in Bt-producing transgrafted poplar and cotton plants [9,10,11].

### 3.3. Cry1Ab Protein Detection in Transgrafted Scion Seeds

Our subsequent investigation focused on the presence of Cry1Ab protein in seeds obtained from the transgrafted plants using LC–MS/MS analysis. For this study, we utilized the Bt39 and Bt44 plants as rootstocks for transgrafting. We subjected the total proteins extracted from the dry seeds of WT, Bt39, and Bt44 plants, along with seeds from six independent transgrafted plants (WT/Bt39-1–3 and WT/Bt44-1–3), to immunoprecipitation using the anti-Cry1Ab antibody. Consistent with expectations, we detected Cry1Ab protein in the Bt39 and Bt44 seeds, but not in the WT seeds. The number of detected peptides in the seed samples was fewer than those observed in the Bt44 leaf samples (Table 1). This observation is in line with previous research indicating that the activity of the CaMV 35S promoter is relatively weak in dry seeds when compared with leaf and root tissues [17], suggesting that the expression level of the *Cry1Ab* gene would be low in the dry seeds of Bt39 and Bt44 plants. In stark contrast, we could not detect Cry1Ab protein in the seeds obtained from any of the total of six transgrafted plants (Table 1).

### 3.4. Proteomic Analysis of Cry1Ab Peptides in Transgenic and Transgrafted Plants

We mapped the peptides identified by LC–MS/MS analysis onto the Cry1Ab amino acid sequence (Figure 2 and Figure 3). Peptides detected from the samples of *Bt* transformants, specifically leaves of Bt44 plants and seeds of Bt39 and Bt44 plants, covered a broad region of the Cry1Ab amino acid sequence (Figure 2). In contrast, the peptides detected from the transgrafted samples concentrated in a few specific regions, particularly in the region identified as peptide 1 (amino acid residues 199 to 209) and peptide 3 (amino acid residues 513 to 522) (Figure 3). We consistently detected these two peptide regions in the non-GM scion leaves of transgrafted plants. While peptide 2 (amino acid residues 266 to 281) was the most abundantly detected in the transgenic samples (Figure 2), we did not detect peptide 2 in the transgrafted samples (Figure 3). The specific form of translocated Cry1Ab protein within scion tissues, including its molecular mass and post-translational modifications, is currently unknown. Therefore, further studies may be necessary to fully explain the observed differences in the LC–MS/MS profiles of detected peptides between the transformant and transgrafted samples.

### 3.5. Co-Identification of Cry1Ab-Interacting Tobacco Proteins

The immunoprecipitated proteins, isolated using an anti-Cry1Ab antibody, included the Cry1Ab protein along with coimmunoprecipitated tobacco proteins. Subsequently, we sought to identify tobacco proteins that might interact with the Cry1Ab protein. To accomplish this, we identified proteins that met the following criteria: tobacco proteins that were present in Cry1Ab-positive leaf samples (Bt44-1–4, WT/Bt44-1, 2, 3, and 5) but not in Cry1Ab-negative leaf samples (WT1 and WT/Bt44-4) (Table 1). This yielded three candidates (Table 2). As Bt proteins lack known plant metabolic activity, commercial Bt products are presumed not to interact with host proteins [18]. Therefore, further research is needed to confirm interactions between these three tobacco proteins and Cry1Ab.

## 4. Discussion

Bt proteins, when produced in transgenic plants, have long been a valuable tool for insect management. Cry1Ab is an insecticide that effectively targets lepidopteran species [19]. Our findings indicate that the Cry1Ab protein translocated from the GM rootstock to the leaves of non-GM scions, which may offer a new strategy for the management of herbivorous insects. Similar translocation of Bt proteins in transgrafted plants has been previously reported [9,10,11]. Although we did not ascertain whether the Cry1Ab protein exhibited insecticidal properties, the translocated Bt protein in the transgrafted poplar plants demonstrated a lethal impact on the target insect larvae [10]. The detection of Bt protein in the xylem sap of Bt cotton plants offers a plausible explanation for its rootstock-to-scion movement [9]. Conversely, Wang et al., (2012) demonstrated that Bt protein transport in transgrafted poplar occurred pre-dominantly through the phloem [10]. Furthermore, our prior research indicated that the LUC protein can translocate bidirectionally (rootstock to scion and vice versa) [6]. Therefore, NEPs synthesized in GM tissues are likely transported via both xylem and phloem.

Environmental factors such as nitrogen content and planting density are known to influence Bt protein levels [20,21], attributable to decreased protein synthesis and elevated degradation [22]. Our Western blot analysis, which failed to detect Cry1Ab proteins in Bt44 plants (Appendix A), suggests that Cry1Ab protein undergoes frequent degradation in these lines. Furthermore, even after immunoprecipitation-mediated concentration, the number of detected Cry1Ab peptides remained relatively low (Table 1). Despite anticipating a low amount of Cry1Ab protein translocated from the rootstock to the scion, we successfully detected Cry1Ab protein in four of the five transgrafted plants. This result strongly implies that translocation of NEPs in transgrafted plants warrants food safety consideration irrespective of their expression level in the GM parent plants. In the previous studies, cell to cell diffusion of proteins appeared to be limited to proteins smaller than 60 kDa [23,24]. Given that Cry1Ab protein is estimated to be 72.6 kDa, it is possible that the detected Cry1Ab protein in the non-GM scion of the transgrafted plants may have undergone partial degradation. This hypothesis may explain the absence of peptide 2 (amino acid residues 266 to 281) in the transgrafted samples (Figure 3).

Long-distance phloem transport of the floral signaling protein FLOWERING LOCUS T (FT) is well-established [25]. This transport is facilitated by SODIUM POTASSIUM ROOT DEFECTIVE 1 (NaKR1), a heavy-metal-associated domain-containing protein that interacts with FT [26], suggesting that the protein interactions with specific translocation facilitators are essential for long-distance protein delivery through the phloem. Given that it is improbable that non-native proteins would interact with specific translocation facilitators, there appears to be a more general route for the translocation of proteins such as Cry1Ab. In line with this, neither LUC nor GFP was detected in tomato fruits from non-GM scions grafted onto GM rootstocks expressing these proteins [7]. In the transgrafted walnut plants, β-glucuronidase and neomycin phosphotransferase II proteins produced in the GM rootstocks were absent in the seeds obtained from non-GM scions [27]. Furthermore, tobacco seeds from non-GM scions grafted onto *LUC*-expressing GM rootstocks exhibited no LUC activity [4]. These collective findings indicate that foreign gene products from GM rootstocks can translocate across the graft junction, but their distribution appears limited to relatively short distances.

The regulatory framework for foods derived from transgrafted plants remains unexplored [28]. The potential translocation of NEPs into the non-GM scions underscores the necessity of conducting safety assessments of the foods from transgrafted plants. In the United States and Japan, risks associated with the GM foods are assessed through a product-based policy. The absence of NEPs and transgenes in the foods from transgrafted plants suggests the potential for their eventual classification as non-GM foods in such countries. In this regard, an insecticidal Bt protein produced in the GM tobacco rootstocks is translocated into the foliar tissues of non-GM scions but not into the seeds, which facilitate the commercialization of edible components as non-GM food products. Given that the Cry1Ab protein level was relatively low in the GM rootstocks in this study, further investigation is warranted to determine whether the translocation of Bt proteins into the seeds occurs when the non-GM scion is grafted onto the GM rootstocks exhibiting a high level of Bt protein accumulation.

## Figures and Tables

**Figure 1 biotech-14-00064-f001:**
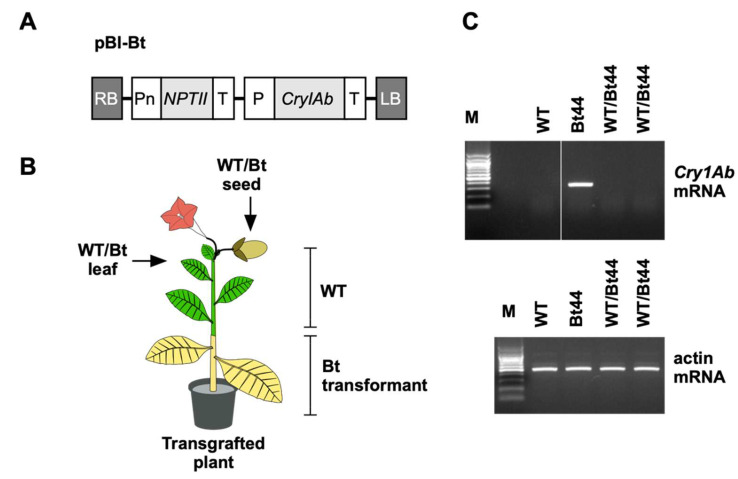
Experimental setup for transgrafted plant production and detection of Cry1Ab mRNA in non-GM scions. (**A**) Schematic representation of the T-DNA region within the pBI-Bt binary vector. RB, right border; Pn, promoter of the nopaline synthase gene; NPTII, neomycin phosphotransferase II; T, terminator of the nopaline synthase gene; P, cauliflower mosaic virus 35S promoter; LB, left border. (**B**) Diagram illustrating the architecture of transgrafted plants. Leaves and seeds were harvested specifically from the non-GM scion portions of these plants. (**C**) Detection of Cry1Ab transcripts via RT-PCR. Total RNA was isolated from leaves of both non-grafted WT and Bt44 plants. Additionally, total RNA was prepared from scion leaves of two independent transgrafted plants (WT/Bt44). The presence of Cry1Ab transcripts was indicated by the generation of 356 bp fragments. As a quantitative loading control, 499 bp fragments were amplified from tobacco actin transcripts.

**Figure 2 biotech-14-00064-f002:**
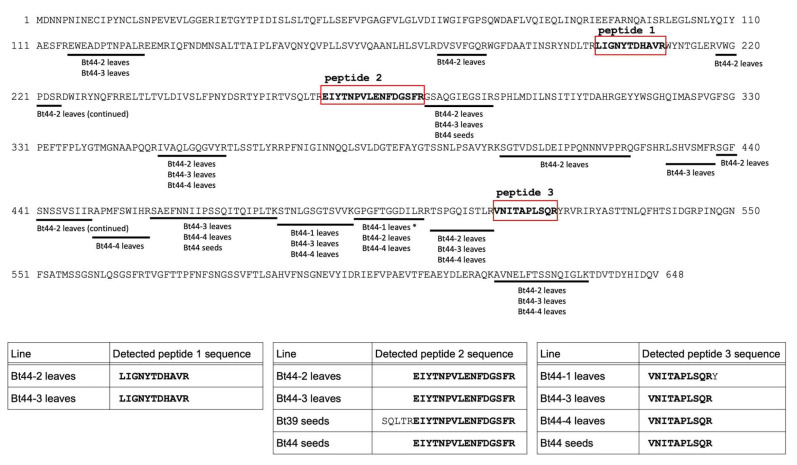
Identification of Cry1Ab peptides in Bt transformants through proteomic analysis. The names of the transgenic lines and the parts of each sample are consistent with the information in Table 1. Peptide fragments identified in each sample are illustrated by black bars positioned beneath the full amino acid sequence of the Cry1Ab protein. Note that identical peptide sequences, even if repeatedly detected, are counted as a single instance, and all unique peptide sequences are displayed together. Three amino acid sequences (peptide 1, 2 and 3) are repeatedly detected in four or more samples of Bt transformants (Figure 2) or non-GM scion of the transgrafted plants (Figure 3). These three sequences are highlighted within red boxes. These peptide sequences are designated as follows: Peptide 1 (LIGNYTDHAVR, corresponding to amino acid positions 199–209), Peptide 2 (EIYTNPVLENFDGSFR, amino acid positions 276–281), and Peptide 3 (VNITAPLSQR, amino acid positions 513–522). Tables within the figure present the detected sequences for Peptides 1, 2, and 3 for each analyzed sample. An asterisk with amino acid positions 491–501 indicates that the peptide sequence of Bt44-1 leaves is “GPGFTGGDILRR”, other peptide sequences (Bt44-2 leaves and Bt44-4 leaves) are “GPGFTGGDILR”.

**Figure 3 biotech-14-00064-f003:**
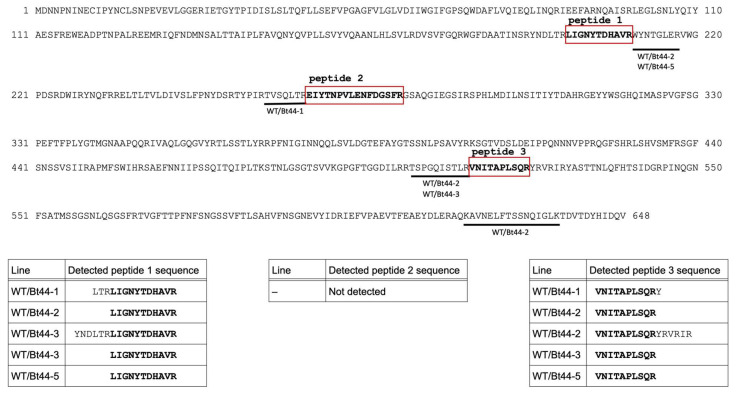
Identification of Cry1Ab peptides in non-GM leaves of transgrafted plants (WT/Bt44-1 to WT/Bt44-5) through proteomic analysis. The names of the transgrafted plants are consistent with the information in Table 1. Three peptide fragments (peptide 1, 2 and 3) are boxed, as stated in the legend of Figure 2. Other peptide fragments identified in each sample are illustrated by black bars positioned beneath the full amino acid sequence of the Cry1Ab protein. Note that identical peptide sequences, even if repeatedly detected, are counted as a single instance, and all unique peptide sequences are displayed together. Tables within the figure present the detected sequences for Peptides 1, 2, and 3 for each analyzed sample. Cry1Ab peptides were detected in leaf samples derived from WT/Bt44-1 to WT/Bt44-3 and WT/Bt44-5 scions. Remarkably, Cry1Ab peptides were not detected in leaves of the WT/Bt44-4 scion nor in any seeds produced from the transgrafted scions.

**Table 1 biotech-14-00064-t001:** Summary of Cry1Ab peptides detected via proteomic analysis and their corresponding coverage of the full Cry1Ab amino acid sequence.

Line	Part	Peptides ^a^	Coverage (%)	−10logP ^b^
WT1 ^c^	Leaves	n.d. ^d^	–	–
WT2	Seeds	n.d.	–	–
WT3	Seeds	n.d.	–	–
Bt39	Seeds	1	2	50.02
WT/Bt39-1	Seeds	n.d.	–	–
WT/Bt39-2	Seeds	n.d.	–	–
WT/Bt39-3	Seeds	n.d.	–	–
Bt44-1	Leaves	3	5	56.33
Bt44-2	Leaves	13	24	106.20
Bt44-3	Leaves	11	21	84.91
Bt44-4	Leaves	8	15	78.96
Bt44	Seeds	4	9	43.49
WT/Bt44-1	Leaves	3	4	47.16
WT/Bt44-2	Leaves	6	9	58.19
WT/Bt44-3	Leaves	4	6	47.43
WT/Bt44-4	Leaves	n.d.	–	–
WT/Bt44-5	Leaves	3	4	40.14
WT/Bt44-6	Seeds	n.d.	–	–
WT/Bt44-7	Seeds	n.d.	–	–
WT/Bt44-8	Seeds	n.d.	–	–

^a^ Number of unique peptides. Even if these unique peptides are repeatedly detected, they are counted as single instance. ^b^ The statistical significance of a detected peptide sequence being annotated to the Cry1Ab protein is expressed as −10logP; for example, a *p*-value of 0.001 is presented as −10logP = 30. ^c^ The numerical suffixes following each tobacco plant line indicate individual, independent plant samples. For instance, “WT1, WT2, WT3” denotes three distinct WT tobacco plants subjected to the analysis. ^d^ “n.d.” signifies not detected.

**Table 2 biotech-14-00064-t002:** List of tobacco proteins coimmunoprecipitated with Cry1Ab protein from leaf samples, excluding the WT/Bt44-4 line.

Uniprot Accession	Description
A0A1S3YM01	40S ribosomal protein S18
A0A1S4B7N5	Catechol oxidase (EC 1.10.3.1)
A0A1S4CEW8	Chlorophyll a-b binding protein, chloroplastic

## Data Availability

The original contributions presented in this study are included in the article/Appendix A. Further inquiries can be directed to the corresponding author(s).

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
