# Peer review of "Translocation of Insecticidal Bt Protein in Transgrafted Plants"

_biotech, 2025, doi:10.3390/biotech14030064_

Round 1
Reviewer 1 Report
Comments and Suggestions for Authors
This study investigates the translocation of the insecticidal protein Cry1Ab (a Bt protein) from genetically modified (GM) tobacco rootstocks to non-GM scions in transgrafted plants. The study addresses a critical regulatory and safety issue in transgrafting technology. The key findings are that Cry1Ab protein is translocated into scion leaves but not into seeds, which not only supports the potential for commercializing transgrafted crops as non-GM food products but also opens up new avenues for the future of agriculture. The novelty is incremental, as similar translocation phenomena have been reported in cotton and poplar. However, the focus on edible tissues (seeds) adds value.
Suggestions for Improvement
1. Experimental design and methodology:
- while 11 transgrafted plants were analyzed, only six were used for seed analysis. Consider discussing whether this sample size is statistically sufficient,
- the study uses WT and GM controls effectively, but including additional scion types or rootstock varieties could strengthen generalizability. Please comment on this point.
- sample size for seed analysis (6 transgrafted plants) may be insufficient to generalize the absence of translocation.
2. Results and Discussion:
- the study mentions low levels of Cry1Ab protein. Could the authors provide quantitative data (e.g., ng/mg tissue) to support this?
- the discussion touches on xylem and phloem transport. It would be valuable to elaborate on why specific peptides (e.g., peptide 2) are absent in transgrafted samples—could this relate to degradation or selective transport?
- the manuscript suggests that the absence of the Bt protein in seeds could significantly facilitate the commercialization of transgrafted crops. However, it's important to note that regulatory frameworks vary globally. A brief discussion of how these findings align with EU, US, or Japanese regulations would be helpful to understand the potential impact on different markets.
- the selective detection of specific peptides (e.g., peptides 1 and 3 but not 2) in transgrafted samples is intriguing but not fully explained. Could this reflect degradation, selective transport, or detection limits?
3. Figures and tables
- figures 2 & 3 - these are informative but could benefit from clearer legends and labeling of peptide regions.
- table 1 - consider adding a column for tissue distance from graft junction to correlate with peptide detection.
4. Language and clarity
- the manuscript is generally well-written, but some sections, such as the methods, could be simplified or broken into subsections to improve readability and comprehension.
- line 29: “significantly facilitate” → consider “may facilitate” to avoid overstatement.
- line 199: “As anticipated” → this implies bias; consider rephrasing to “Consistent with expectations”.
- ensure all abbreviations are defined at first use (e.g., NEP, LC-MS/MS).
Author Response
The revised parts of the main text are indicated in red. In addition to the points noted by the reviewers, the following revisions have been made:
- We deleted the keyword "new plant breeding technology."
- The list of abbreviations has been updated. Additionally, the abbreviations for EDTA, TEAB, and TCEP have been replaced with their full spellings.
- The format of the references has been standardized.
Reviewer 1
Comments 1: - while 11 transgrafted plants were analyzed, only six were used for seed analysis. Consider discussing whether this sample size is statistically sufficient,
Response 1: Hierarchical cluster analysis of the proteomic data of immunoprecipitated seed samples showed that the Bt 44/39 seed samples were clearly separated from the seed samples of WT and WT/Bt transgrafted plants. This indicates that WT seed samples and WT/Bt transgrafted seed samples shared an similar proteomic profiles.
Figure. Hierarchical cluster analysis of the proteomic data of immunoprecipitated seed samples.
Comments 2: - the study uses WT and GM controls effectively, but including additional scion types or rootstock varieties could strengthen generalizability. Please comment on this point.
Response 2: Using scions and rootstocks with different characteristics can yield more generalized results; however, the results also become more complex as the number of examined factors increases. In this study, therefore, we analyzed only Bt-overexpressing rootstocks and non-GM scions to investigate the transfer of proteins derived from foreign genes to the scions. Results obtained with different genetically modified plants have been reported in our previous studies and other papers (lines 307-312).
Comments 3: - sample size for seed analysis (6 transgrafted plants) may be insufficient to generalize the absence of translocation.
Response 3: Based on the detection rate in transgrafted scion leaves (four out of five, Table 1), all six transgrafted seed samples did not show any translocation of Cry1Ab protein from the rootstock. As reported in our previous studies, luciferase activity was not detected in seeds obtained from non-GM scions grafted onto tobacco rootstocks producing luciferase. Similarly, no luciferase and GFP were detected in the fruits of non-GM tomatoes grafted onto GM tomato rootstocks expressing luciferase and GFP. Similar results have been observed in walnut seeds (see the additional reference in the main text by Haroldsen et al., 2012). Therefore, we conclude that newly expressed proteins in the GM rootstock are not transferred to the seeds of non-GM scions in most cases. We added some description about this point in lines 307-314.
Comments 4: - the study mentions low levels of Cry1Ab protein. Could the authors provide quantitative data (e.g., ng/mg tissue) to support this?
Response 4: Quantitative data are not available. We measured the Cry1Ab proteins using an LC–MS/MS analysis of immunoprecipitation-enriched fractions, this method would be one of the most sensitive detection methods that are currently available.
In addition, some peptides such as peptide 2 are lack in the transgrafted leaf samples, which suggest the partial degradation of Cry1Ab proteins in the non-GM scion leaves. In such case, accurate quantification using ELISA may be difficult.
Comments 5: - the discussion touches on xylem and phloem transport. It would be valuable to elaborate on why specific peptides (e.g., peptide 2) are absent in transgrafted samples—could this relate to degradation or selective transport?
Response 5: As mentioned in Comment 4, there is a possibility of fragmentation of Cry1Ab protein during translocation. We have added a report indicating that there is a molecular weight limit for transport in the phloem and included this in the discussion (lines 295-300). Additionally, it has been reported that Bt proteins can be transported through both the xylem and phloem.
Comments 6: - the manuscript suggests that the absence of the Bt protein in seeds could significantly facilitate the commercialization of transgrafted crops. However, it's important to note that regulatory frameworks vary globally. A brief discussion of how these findings align with EU, US, or Japanese regulations would be helpful to understand the potential impact on different markets.
Response 6: We added some description about the present status for GM regulartory policies in United States and Japan in the main text (lines 315-321). The GM regulatory panel in EU is now under consideration how they handle the new products obtained using new plant breeding technology.
Comments 7: - the selective detection of specific peptides (e.g., peptides 1 and 3 but not 2) in transgrafted samples is intriguing but not fully explained. Could this reflect degradation, selective transport, or detection limits?
Response 7: As you pointed out, the possible relationship between the lack of peptide 2 and speculated partial degradation of Cry1Ab protein detected in the non-GM scion leaves of the transgrafted plants were mentioned in the main text (lines 295–300).
Comments 8: - figures 2 & 3 - these are informative but could benefit from clearer legends and labeling of peptide regions.
Response 8: We revised the legends for Figures 2 and 3. Enclosing the detected peptide sites in frames makes identifying the start and end points of consecutive peptide fragments difficult. Therefore, we left the black bars as they are in the 1st submitted manuscript.
Comments 9: - table 1 - consider adding a column for tissue distance from graft junction to correlate with peptide detection.
Response 9: The distance from the graft junction is listed in Table S2.
Comments 10: - the manuscript is generally well-written, but some sections, such as the methods, could be simplified or broken into subsections to improve readability and comprehension.
Response 10: The LC–MS/MS section has been divided into two subsections.
Comments 11: - line 29: “significantly facilitate” → consider “may facilitate” to avoid overstatement.
Response 11: We corrected it at line 30.
Comments 12: - line 199: “As anticipated” → this implies bias; consider rephrasing to “Consistent with expectations”.
Response 12: We corrected it at line 214.
Comments 13: - ensure all abbreviations are defined at first use (e.g., NEP, LC-MS/MS).
Response 13: We corrected it.

Reviewer 2 Report
Comments and Suggestions for Authors
The topic is of key importance for agriculture, as the article addresses a critical regulatory and biosafety issue in transplant technology. The combination of molecular detection and proteomics is convincing, and the absence of Cry1Ab in seeds is vital for the classification as non-modifying. The work deserves publication after minor edits to improve the interpretation of the data and contextual discussion.
Comments:
1. The introduction should tell more about B. thuringiensis as the source of Cry1Ab. From which subspecies do the gene and, accordingly, the protein originate? In general, the introduction should start with the problems in agriculture, some statistics on pest damage (especially for tobacco), and the history of successful transgenic plants (at least 30 years). The introduction and discussion are too short; both should be significantly expanded.
2. The authors should do some more precise quantification of proteins, e.g., relative or absolute Cry1Ab protein amounts (e.g., ng/g FW) in scion leaves and rootstock tissues to contextualize the significance of translocation.
3. The authors should expand the discussion on the possible transport pathways (xylem, phloem, or protein-specific carriers) and relate their findings to the current knowledge on protein mobility.
4. In the "Introduction", the authors promise a "discussion on the potential risks of NEP translocation into edible transgrafted plant portions", but this discussion is limited to one sentence regarding tomatoes, and the reader expects much more. Moreover, the model system is tobacco; there are no parts that can be used for food. How can this system be applied to other plants, especially edible crops?
Author Response
The revised parts of the main text are indicated in red. In addition to the points noted by the reviewers, the following revisions have been made:
- We deleted the keyword "new plant breeding technology."
- The list of abbreviations has been updated. Additionally, the abbreviations for EDTA, TEAB, and TCEP have been replaced with their full spellings.
- The format of the references has been standardized.
Reviewer 2
Comments 1: The introduction should tell more about B. thuringiensis as the source of Cry1Ab. From which subspecies do the gene and, accordingly, the protein originate? In general, the introduction should start with the problems in agriculture, some statistics on pest damage (especially for tobacco), and the history of successful transgenic plants (at least 30 years). The introduction and discussion are too short; both should be significantly expanded.
Response 1: The Cry1Ab used in this study is derived from Bacillus thuringiensis subsp. kurstaki. The text has been revised and expanded to include lines 63-67. The introduction has been expanded to include information on how Bt proteins have helped solve agricultural problems and increase farmers' incomes (lines 45–46). A brief overview of the history of genetically modified (GM) foods and their regulation has been added to the introduction (lines 49–53). This study uses tobacco as a model plant, and our aim does not include the use of Bt proteins in tobacco cultivation. This point has been clarified in the text (lines 63–64). Information on Cry1Ab's approval status for use in GM foods has been added (lines 65–67).
Comments 2: The authors should do some more precise quantification of proteins, e.g., relative or absolute Cry1Ab protein amounts (e.g., ng/g FW) in scion leaves and rootstock tissues to contextualize the significance of translocation.
Response 2: Quantitative data are not available. We measured the Cry1Ab proteins using an LC-MS/MS analysis of immunoprecipitation-enriched fractions, this method would be one of the most sensitive detection methods that are currently available.
In addition, some peptides such as peptide 2 are lack in the transgrafted leaf samples, which suggest the partial degradation of Cry1Ab proteins in the non-GM scion leaves. In such case, accurate quantification using ELISA may be difficult.
Comments 3: The authors should expand the discussion on the possible transport pathways (xylem, phloem, or protein-specific carriers) and relate their findings to the current knowledge on protein mobility.
Response 3: The mechanism by which substances are transported from the rootstock to the scion is described using FT proteins as an example. Additionally, information has been added to improve understanding of the transport of introduced gene products (lines 301-307).
Comments 4: In the "Introduction", the authors promise a "discussion on the potential risks of NEP translocation into edible transgrafted plant portions", but this discussion is limited to one sentence regarding tomatoes, and the reader expects much more. Moreover, the model system is tobacco; there are no parts that can be used for food. How can this system be applied to other plants, especially edible crops?
Response 4: In addition to the tomato example mentioned in the introduction, several examples have been added (lines 55-60). Considerations regarding the future use of foods obtained through transgrafting have also been included (lines 315–321).
